# Development and Validation of Case-Finding Algorithms for Digestive Cancer in the Spanish Healthcare Database BIFAP

**DOI:** 10.3390/jcm13020361

**Published:** 2024-01-09

**Authors:** Encarnación Fernández-Antón, Antonio Rodríguez-Miguel, Miguel Gil, Amelia Castellano-López, Francisco J. de Abajo

**Affiliations:** 1Clinical Pharmacology Unit, University Hospital “Príncipe de Asturias”, 28805 Madrid, Spain; 2Department of Biomedical Sciences (Pharmacology), University of Alcalá (IRYCIS), 28805 Madrid, Spain; 3BIFAP (Base de datos para la Investigación Farmacoepidemiológica en el Ámbito Público), Division of Pharmacoepidemiology and Pharmacovigilance, Spanish Agency for Medicines and Medical Devices (AEMPS), 28022 Madrid, Spain; 4Department of Gastroenterology, University Hospital “Príncipe de Asturias”, 28805 Madrid, Spain

**Keywords:** electronic health databases, digestive cancer, validation study, electronic health records, pharmacoepidemiology

## Abstract

Background: electronic health records (EHRs) are helpful tools in epidemiology despite not being primarily collected for research. In Spain, primary care physicians play a central role and manage patients even in specialized care. All of this introduces variability that may lead to diagnostic inconsistencies. Therefore, data validation studies are crucial, so we aimed to develop and validate case-finding algorithms for digestive cancer in the primary care database BIFAP. Methods: from 2001 to 2019, subjects aged 40–89 without a cancer history were included. Case-finding algorithms using diagnostic codes and text-mining were built. We randomly sampled, clustered, and manually reviewed 816 EHRs. Then, positive predictive values (PPVs) and 95% confidence intervals (95% CIs) for each cancer were computed. Age and sex standardized incidence rates (SIRs) were compared with those reported by the National Cancer Registry (REDECAN). Results: we identified 95,672 potential cases. After validation, the PPV (95% CI) for hepato-biliary cancer was 87.6% (81.8–93.4), for esophageal cancer, it was 96.2% (93.1–99.2), for pancreatic cancer, it was 89.4% (84.5–94.3), for gastric cancer, it was 92.5% (88.3–96.6), and for colorectal cancer, it was 95.2% (92.1–98.4). The SIRs were comparable to those reported by the REDECAN. Conclusions: the case-finding algorithms demonstrated high performance, supporting BIFAP as a suitable source of information to conduct epidemiologic studies of digestive cancer.

## 1. Introduction

Despite remarkable advances in diagnosis and treatment, cancer remains a major challenge for public health. In Spain, the last large epidemiologic study published showed that cancer is the second cause of death after cardiovascular diseases [1]. More specifically, digestive cancer (including esophagus, gastric, hepato-biliary, pancreatic, and colorectal) is among the top 10 in number of deaths, accounting for 36% of all cancer-related deaths [1]. Further, colorectal cancer is the third most common type of cancer among men and the second among women, resulting in the highest cancer incidence when both sexes are considered [1].

Electronic health records (EHRs) stored in automated databases have long been used as a key source of information in pharmacoepidemiology [2]. EHRs gather longitudinal health data such as signs and symptoms, procedures, or drug prescriptions, among others, recorded by physicians as part of their daily routine. However, EHRs are intended for medical care rather than research [3], so for the latter purpose, researchers must address issues such as diagnostic inconsistency or ambiguity, or the analysis of unstructured data to ensure the validity of the outcomes of interest [4,5,6]. Furthermore, when databases are not linked with an external source, such as specialized care or cancer and mortality registries, the validity of the outcome relies exclusively on the records maintained by the physician [7], which may lead to misclassification in specific environments. In Spain, primary care physicians (PCPs) are the gatekeepers to specialized care, and patients eventually visit PCPs to monitor their diseases and health problems, even if they were diagnosed with specialized care. BIFAP (“Base de datos para la Investigación Farmacoepidemio-lógica en el Ámbito Público”; www.bifap.org, accessed on 4 December 2023) is a Spanish database mainly comprised of EHRs from primary care, so the study of outcomes from specialized care requires prior validation.

Case-finding algorithms based on codes from medical dictionaries, in some cases, also enriched with other supporting information from clinical notes in free text, are commonly used for mining electronic databases [8,9]. In order to ensure the validity and reproducibility of these algorithms, it is important that authors declare the code lists used to define the outcomes and assess the accuracy of the algorithms via comparison against a gold standard, such as a manual review of the EHRs. Moreover, mining large electronic databases could return a huge number of potential outcomes, rendering the manual review of EHRs highly impractical; therefore, designing efficient validation strategies is also crucial.

In the context of a previous study about chemoprevention of colorectal cancer (CRC) in the BIFAP, our research group developed a validation strategy based on the clustering of potential cases according to the recorded information and the manual review of a random sample of the EHRs from each cluster. Finally, the results from the samples were extrapolated to the whole cluster when a high positive predictive value (PPV) was reached [10].

In the present study, we aimed to build case-finding algorithms to identify the incident cases of digestive cancer, including esophageal, gastric, hepato-biliary, pancreatic, and colorectal cancer, tailored to the characteristics of the BIFAP, and to design an efficient validation strategy to reduce the requirements of time and human resources. Finally, as an external validation method, we compared the age and sex standardized incidence rates (SIRs) in the BIFAP with those reported by the Spanish Network of Cancer Registries (REDECAN) to ensure the suitability of the database for the research of an outcome like digestive cancer.

## 2. Methods

### 2.1. Source of Information

This study was conducted using the BIFAP, a longitudinal population-based electronic database funded by the Spanish Agency for Medicines and Medical Devices (AEMPS). The BIFAP comprises pseudonymized EHRs from patients attended by PCPs and pediatricians from the Spanish National Health System (NHS) as part of their routine activities. Information in the BIFAP includes demographic data, diagnoses, specialist referrals, clinical notes as free text, drug prescriptions, and other relevant health data (e.g., lifestyle habits or laboratory results, among others). Diagnoses at hospital discharge are also being progressively linked but are currently only available for half of the population in the last five years. Conversely, drugs prescribed or administered during hospital stays and inpatient clinical notes are not available [11,12].

In 2022, the NHS covered almost the entire Spanish population (96.5%) [13], and the BIFAP contained health information from ten out of the seventeen autonomous regions of Spain, which represented 35.9% of the Spanish population and 91.6% of the participating regions. The BIFAP is updated every six months. In the 2023 version, the BIFAP was contributed to by 14,810 PCPs and pediatricians, gathering 20.8 million EHRs, with an average follow-up of nine years per subject. The age and sex distribution of the population in the BIFAP was comparable to that of the whole country [11]. Personal data is doubly de-identified, so researchers have no access to the personal data of patients or their physicians.

Diseases, other health problems, and interventions are coded using two dictionaries: the International Classification of Primary Care version 2 (ICPC-2) and the International Classification of Diseases version 9 (ICD-9). Both coding systems differ in granularity, with the ICPC having less granularity than the ICD-9 (1300 codes vs. 13,000 codes, respectively). PCPs record the episodes of interest as an incident diagnosis (ID), a clinical problem list (CPL), and/or a personal history (PH). Software tools assist PCPs in this process by offering the type of codes and disease descriptions that would best match the episode to record. However, PCPs can modify these descriptors to better detail the episodes, and these modifications are stored for further use, increasing the variability of diagnosis mapped to the different ICPC-2 codes. The ICPC-2 is most commonly used in primary care and holds most of that variability; therefore, to reduce it, staff from the BIFAP have created an improved version named ICPC-BIFAP that increases the granularity of the ICPC-2 dictionary and covers more than 90% of the total number of diagnoses included in the BIFAP database [11]. Additional information registered by the PCP (clinical notes as free text, laboratory tests, prescriptions, etc.) linked to the diagnosis is also available for its use in the BIFAP.

### 2.2. Study Design and Development of the Case-Finding Algorithms

This validation study was framed in a cohort study aimed at assessing the chemoprotective effect of antiplatelet drugs on digestive cancer. From 1 January 2001 to 31 December 2019, we included all subjects aged 40–89 years, of any gender, with at least 1 year of previous registry with their PCP, and without a prior history of any type of cancer (excluding non-melanoma skin cancer).

For the validation study, we expanded the methodology developed in a previous study about colorectal cancer in the BIFAP described elsewhere [10] to encompass other types of digestive cancer, including esophageal, gastric, hepato-biliary (hepatocellular and cholangiocarcinoma), pancreatic, and colorectal cancers. To that end, we built case-finding algorithms fitted to each type of digestive cancer, which included the following ICPC-BIFAP and ICD-9 codes: for esophageal cancer: D77.2/8, 150–150.9, 230.1; for gastric cancer: D74.1, D75.1003, 151–151.9, 230.2; for pancreatic cancer, D76.1/4/5, 157–157.9, M8154/3; for hepatobiliary cancer: D77.4/5, 155–155.2, 156–156.9, 230.8, M8160/3, M8161/3, M8170/3, M8180/3, M8970/3, M8970/6; and for colorectal cancer: D75.1/4/5/6/999/1004/1005, 153.0–153.9, 154–154.8, 230.3–230.4. The algorithms were additionally enriched with text mining in either the descriptors associated with the ID, PH, and CPL fields or in the free-text clinical notes (more is detailed in Appendix A). We then deployed each algorithm in the BIFAP to identify all the potential incident cases of digestive cancer throughout the study period. Neuroendocrine and genetically based digestive cancers (e.g., Lynch syndrome) were excluded.

### 2.3. Validation Strategy for Digestive Cancer Diagnoses

All potential cases were grouped according to the type of cancer and were stratified depending on the information completeness: the first stratum included a digestive cancer diagnosis (as a diagnostic code or as a descriptor in the free text associated with a code different to digestive cancer), clinical notes as free text, and supporting information. Regarding the latter, we considered the following: stage, location, surgery, chemo or radiotherapy, diagnostic codes at hospital discharge or specialist reports, and confirmation via histopathology and/or diagnostic imaging or colonoscopy. The second stratum included a digestive cancer diagnosis and clinical notes as free text, and the third included only a digestive cancer diagnosis.

We randomly sampled 100 EHRs from each group that were manually reviewed by two independent researchers according to the following schedule: first, they jointly reviewed a sample of EHRs to settle any differences and reach a consensus; second, both independently worked on the same EHRs until a kappa coefficient of agreement was equal or greater to 0.9; and finally, the remaining sample of EHRs was randomly allocated to each researcher and was independently reviewed. Each reviewer classified the potential cases into (1) valid case, (2) non-valid case (including other relative diagnoses, genetic or neuroendocrine digestive cancer, screening activities, other non-digestive cancer, a benign tumor, or a prevalent digestive cancer), (3) inconclusive case (when ambiguous information was found e.g., colon polyps vs. colorectal cancer), and (4) unsupported case (when no additional information to confirm the case was found). The reviewers were blinded to the drug prescriptions to avoid differential misclassification bias of the disease. The unclassified cases and disagreements were arbitrated by a validation committee composed of the leader of the research group, a gastroenterologist, a senior epidemiologist from the BIFAP, and the two reviewers. The validation committee studied each case and assigned the final classification. As the majority of cases fell within the stratum with the most complete information, we randomly sampled up to 40 additional EHRs from the remaining strata in order to increase the accuracy of the estimates.

Thereafter, the positive predictive value (PPV) with their 95% confidence interval (95% CI) was calculated for each stratum, cancer group, and digestive cancer overall.

### 2.4. Fine-Tuning of the Case-Finding Algorithms

We re-examined the EHRs from the false positives to detect the patterns that could feed the case-finding algorithms and improve their performance in a second iteration. Our case-finding algorithms hierarchically searched among a selection of diagnostic codes but sometimes led to random classifications when two types of cancer co-occurred. As an example, a case of “rectal cancer with liver metastasis” was detected with the algorithm as rectal and liver cancer but were randomly classified in either group. To solve that, we changed the hierarchy of the diagnostic codes accordingly. Similarly, free-text searches led the algorithm to misclassify some cases of colorectal cancer located in the hepatic angle into hepato-biliary cancer, so we refined the free-text mining technique to avoid that. Finally, diagnostic codes at hospital discharge were also included when available (Appendix A).

The validation committee agreed to classify gastroesophageal junction cancer within esophageal cancer and ampulloma within hepato-biliary cancer.

### 2.5. Statistical Analysis

The inter-reviewer level of agreement was estimated through the linear weighted Cohen’s kappa coefficient.

In each random sample, the positive predictive value (PPV) was calculated as the number of valid cases after manual review divided by the total number of potential cases identified with the case-finding algorithms. The PPVs and 95% CIs were computed for each stratum, and by type of cancer as a mean weighted by the proportion of each stratum within the group. Additionally, to rule out the differential quality of records, the PPVs were also stratified according to sex.

For the main analysis, we considered the inconclusive cases non-valid, and the unsupported cases were excluded. However, in a sensitivity analysis, we considered the unsupported cases under two scenarios as valid and non-valid cases.

The crude incidence rates of digestive cancer were calculated according to the type of cancer as the number of incident cases divided by the total time of follow-up. The age and sex standardized incidence rates (SIRs) and 95% CIs by 100,000 person-years were computed using the European population as the reference [14].

Intervals at 95% confidence for the PPVs and SIRs were estimated using a binomial distribution.

All statistical analyses were performed with STATA/MP v.17.0 (Stata Corp LLC, College Station, TX, USA). A *p*-value of <0.05 was considered statistically significant.

### 2.6. Ethics

The study protocol was approved by the Scientific Committee of the BIFAP on 13 August 2020 (protocol number: ISC-AAS-2020-01). The ethics committee of the University Hospital “Príncipe de Asturias” also approved the study protocol on 2 October 2020 and granted a waiver for informed consent, as all the data were pseudonymized, according to the European and Spanish laws on data protection.

## 3. Results

### 3.1. Study Cohort and Identification of Potential Cases of Digestive Cancer

A total of 4,045,411 subjects fulfilling the eligibility criteria were finally included in the study cohort. The algorithms identified 95,672 potential cases, 33.3% of which were detected after text mining based on descriptors not associated with a diagnostic code of digestive cancer. The largest group was colorectal cancer (*n* = 62,787; 65.6%), followed by gastric (*n* = 11,161; 11.7%) and hepato-biliary (*n* = 9418; 9.84%) cancers (Figure 1).

Overall, 84% of cases were classified in the first stratum of information completeness (cancer diagnosis + clinical notes as free text + supporting information), while the remaining two strata (cancer diagnosis + clinical notes as free text and cancer diagnosis) accounted for 10% and 6%, respectively.

### 3.2. Validation of Potential Cases of Digestive Cancer

Within each cancer group, 12 random EHRs were reviewed to reach a consensus, and then, the reviewers independently reviewed 100 random EHRs. After the first 20, we reached a kappa coefficient of 0.91 (*p* < 0.001).

The validation results are shown in Table 1. A total of 56 unsupported cases were excluded from the main analysis: 19 from hepatobiliary, 7 from esophageal, 14 from pancreatic, 7 from gastric, and 9 from colorectal cancer. Before fine-tuning, the case-finding algorithm yielded a weighted mean PPV beyond 80% for all types of cancer, except for hepato-biliary cancer, which was 71.1% (95% CI: 63.5–78.7%) (Appendix A).

The fine-tuned algorithms improved all results from the first iteration. Of note, the PPV (95% CI) for hepato-biliary cancer rose to 87.6% (81.8–93.4%), and up to 92.5% (88.3–96.6%) for gastric cancer (Table 1).

The sensitivity analyses, in which unsupported cases were evaluated under two valid and non-valid scenarios, barely changed these results (Appendix A). Furthermore, the PPVs stratified according to sex remained consistent as expected (Appendix A).

### 3.3. Standardized Incidence Rates of Digestive Cancer in the BIFAP

We compared the age and sex SIRs (95% CI) of all the digestive cancers in the BIFAP with those provided by the REDECAN for 2012 (the latest available). The SIR (95% CI) for hepato-biliary cancer was 7.2 (6.3–8.0) in the BIFAP and 8.6 by the REDECAN; for esophageal cancer, it was 2.8 (2.2–3.3) and 3.5, respectively; for pancreatic cancer, it was 6.3 (5.5–7.1) and 9.4, respectively; for gastric cancer, it was 13.5 (12.3–14.7) and 11.6, respectively; and for colorectal cancer, it was 48.1 (45.8–50.3) and 48.9, respectively.

The SIRs according to gender in both the BIFAP and REDECAN were higher in males than in females among all types of digestive cancers. In females, the SIRs from the BIFAP were more comparable with those of the REDECAN than in males (Figure 2).

## 4. Discussion

In the present study, we were able to develop high-performance algorithms for the identification of the cases of digestive cancer in the BIFAP and efficiently validated them with controlled costs in human resources and time. All in all, our results confirmed that the BIFAP was a reliable source of information to conduct studies using digestive cancer as the outcome of interest.

In Spain, digestive cancer is diagnosed in specialized care, while PCPs are established as the cornerstone of the NHS and manage diseases and health problems of almost all patients within it. Therefore, it would be expected that a primary care database such as BIFAP could capture this reality and contain relevant information about such diseases. In contrast, there is variability in real-world data, especially since PCPs are allowed to modify the descriptors of ICPC-2 codes to adapt it to the particular case better, so the codes alone not always unambiguously identify the disease. In fact, this is a critical issue regarding the validity of the results from pharmacoepidemiologic studies using only code-based case-finding algorithms to define the outcome of interest [15,16]. To tackle this, some authors advocate for the inclusion of free-text mining over descriptors and clinical notes in addition to diagnostic code searches to improve the performance of the case-finding algorithms and reduce misclassification bias [10,16]. In this sense, 33.3% of our cases were detected after mining over descriptors linked to diagnostic codes different from digestive cancer, so otherwise, they would have been missed. Furthermore, 84% of our cases had a diagnostic code and/or descriptors of digestive cancer plus additional information from notes in free text and supporting information, which, in addition, yielded the highest validity.

We obtained overall PPVs higher than 87% for all types of digestive cancer, which is in line with the observations of previous studies using similar databases [17,18]. We fixed a sample threshold PPV of equal or more than 85% to consider a whole group as valid. At this threshold and with a large sample size such as ours, we could be sure that the residual misclassification of the disease would barely impact the measures of association in a pharmacoepidemiologic study. Still, it would also be expected to be non-differential with respect to the exposure, as reviewers were blinded to drug prescriptions. As an example, the main results barely changed in a sensitivity analysis after misclassifying the unsupported cases as valid. This was possible with controlled costs in human resources and time, as from a random sample of 816 EHRs, which were homogeneously clustered and manually reviewed, we were able to extrapolate the sample validation results to the entire pool of 95,672 potential digestive cancer cases identified in the BIFAP. The manual validation process, including all the stages described in the present study, took us almost a year of work. Managing huge sample sizes is common when using large electronic databases, so it is needless to say that the manual review of the whole pool of potential cancers would have been impractical, which highlights the importance of designing an efficient strategy.

Our group performed a validation study of colorectal cancer diagnoses by applying a similar methodology to an older version of the BIFAP (2015) [10]. In that study, we obtained a lower PPV (87.3%), mainly due to the following methodological differences: (1) we considered suitable cases those with just a code in the PH or CPL fields, and these usually were the prevalent cases. In the present study, we initially excluded those from the algorithms, and (2) diagnoses at hospital discharge were unavailable for that period, while in the present study, they served as the gold standard to confirm valid, unsupported, and inconclusive cases.

In a validation study of different types of cancer using a similar primary care database from Catalonia (SIDIAP), a region within Spain not included in the BIFAP, the authors obtained PPVs for esophagus, gastric, colorectal, liver, gallbladder, biliary tract, and pancreas cancers, ranging from 53.3% to 71.9%. They were able to link the cases found in the database with two cancer registries of that region, also allowing them to evaluate the sensitivity of the algorithm. In contrast, they obtained lower PPVs than ours, which could be partially explained by the inclusion of subjects that were 18 to 35 years old, whose PPVs for certain types of digestive cancer were very low. Of note, as the authors state in the limitations of the study, they did not mine the clinical notes in the free text to help distinguish suspicions from actual diagnoses [19]. That technique revealed an important improvement in our algorithms. In another study, the authors linked the cases with a cancer registry, which was also complemented with a manual review of the medical profiles, reporting a similar PPV for colorectal cancer (94%), while for pancreatic cancer, they reported a PPV of 96%, whereas ours was slightly lower (89.4%) [18]. In this line, we observed lower comparability against the REDECAN of our estimated SIRs for pancreatic cancer but also for gastric and hepato-biliary cancer, especially among males. However, it is plausible that these types of cancers are often diagnosed at advanced stages and are associated with very low survival rates, so this may complicate follow-ups by PCPs, and that may result in under-recording.

Overall, the age and sex SIRs obtained in the BIFAP were comparable to those reported by the Network of Cancer Registries in Spain (REDECAN) in 2012 [20]. There is a lack of updated data on the incidence of digestive cancer in Spain after 2007, with the published data beyond that year relying on long-term predictions [1]. We selected 2012 for comparison as it was the most recent estimation, thus with lower error, and the last offering data for the entire country. In that case, the SIRs obtained in the BIFAP beyond 2007 might provide a better and updated estimation of the actual incidence rate of digestive cancer in the Spanish population.

Some limitations should be acknowledged. We could not link our cases with a gold standard as the cancer registry to compute the sensitivity of the case-finding algorithms, so we assessed their performance by means of the PPV. This metric is highly dependent on the prevalence of the disease and implies that, in the absence of a perfect classifier, increasing PPV lowers the sensitivity, albeit the comparability of the SIRs obtained in the BIFAP against the gold standard of cancer registries (REDECAN) provides reassurance.

Some strengths should be highlighted as well. We improved the performance of our algorithms with the use of text-mining techniques and the inclusion of diagnoses at hospital discharge. In addition, we refined the algorithms and outperformed the primary results. Of note, in the absence of updated data, the SIRs estimated in the BIFAP may be more reliable than the predictions based on time series from 1997 to 2007 published so far in Spain.

## 5. Conclusions

In conclusion, our results confirm that the BIFAP is a reliable source of information for the study of the epidemiology of digestive cancer as well as to conduct pharmacoepidemiologic studies on this topic. However, to ensure success, case-finding algorithms, including diagnostic codes and textual information analysis, must be built and validated.

## Figures and Tables

**Figure 1 jcm-13-00361-f001:**
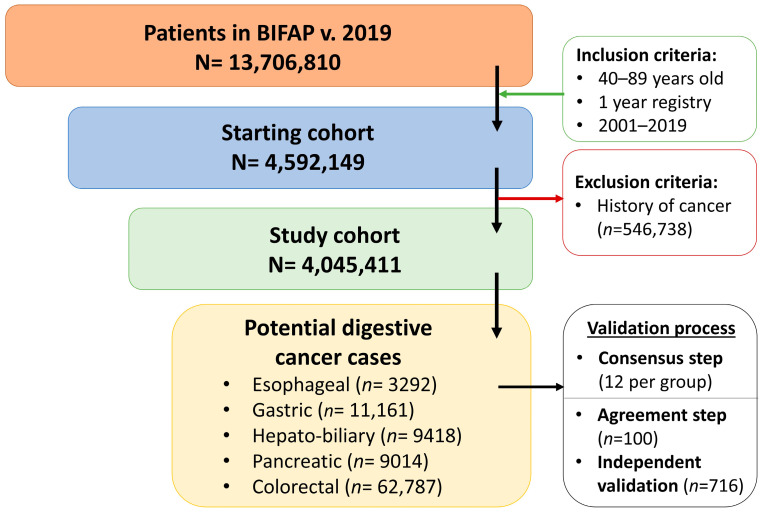
Cohort inception and number of potential cases of digestive cancer identified using the case-finding algorithm in the BIFAP.

**Figure 2 jcm-13-00361-f002:**
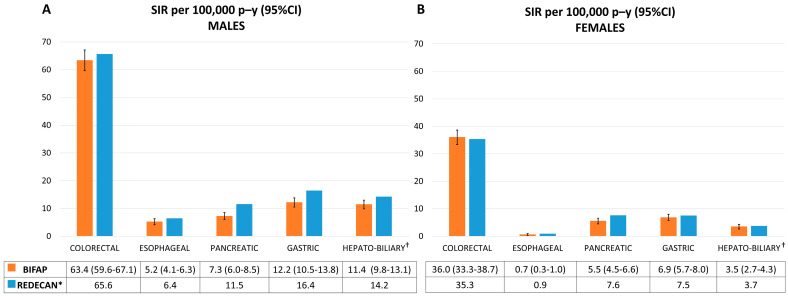
Age and sex standardized incidence rates (SIRs) per 100,000 person-years of digestive cancer in the BIFAP and by the REDECAN from 2012. (**A**): Age standardized incidence rates for males. (**B**): Age standardized incidence rates for females. SIR: standardized incidence rate; p–y: person-years; CI: confidence interval; BIFAP: Base de datos para la Investigación Farmacoepidemiológica en el Ámbito Público; REDECAN: Red Española de Registros de Cáncer. * 95% CIs were not reported. ^†^ SIR for liver cancer. Data for hepato-biliary cancer were not provided.

**Table 1 jcm-13-00361-t001:** Validation results before and after fine-tuning.

Type of Cancer	Information Stratum	PPV (95%CI), %
Hepato-biliary (*n* = 124)	(1) Cancer diagnosis + clinical notes as free text + supporting information	88.5 (77.8–95.3)
	(2) Cancer diagnosis + clinical notes as free text	86.8 (71.9–95.6)
	(3) Cancer diagnosis	80.0 (59.3–93.2)
	(4) Overall weighted mean	**87.6 (81.8–93.4)**
Esophageal (*n* = 152)	(1) Cancer diagnosis + clinical notes as free text + supporting information	96.4 (89.9–99.3)
	(2) Cancer diagnosis + clinical notes as free text	91.9 (78.1–98.3)
	(3) Cancer diagnosis	100
	(4) Overall weighted mean	**96.2 (93.1–99.2)**
Pancreatic (*n* = 152)	(1) Cancer diagnosis + clinical notes as free text + supporting information	88.2 (79.4–94.2)
	(2) Cancer diagnosis + clinical notes as free text	94.7 (82.3–99.4)
	(3) Cancer diagnosis	89.7 (72.6–97.8)
	(4) Overall weighted mean	**89.4 (84.5–94.3)**
Gastric (*n* = 158)	(1) Cancer diagnosis + clinical notes as free text + supporting information	94.0 (86.5–98.0)
	(2) Cancer diagnosis + clinical notes as free text	80.0 (64.4–90.9)
	(3) Cancer diagnosis	94.3 (80.8–99.3)
	(4) Overall weighted mean	**92.5 (88.3–96.6)**
Colorectal (*n* = 174)	(1) Cancer diagnosis + clinical notes as free text + supporting information	96.0 (90.0–98.9)
	(2) Cancer diagnosis + clinical notes as free text	89.5 (75.2–97.1)
	(3) Cancer diagnosis	91.9 (78.1–98.3)
	(4) Overall weighted mean	**95.2 (92.1–98.4)**
Overall (*n* = 760) *	(1) Cancer diagnosis + clinical notes as free text + supporting information	93.0 (90.0–95.2)
	(2) Cancer diagnosis + clinical notes as free text	89.5 (84.3–93.5)
	(3) Cancer diagnosis	91.7 (86.3–95.5)
	(4) Overall weighted mean	**92.4 (90.5–94.3)**

PPV: positive predictive value; CI: confidence interval. * 56 unsupported cases were excluded from the main analysis. The bold emphasis highlights the main PPV result within the cancer groups, accounting for the weighted stratum.

## Data Availability

Data from the BIFAP is restricted to non-profit organizations and independent researchers, so authors are not allowed to share them. However, the data could be available by reasonable request from any organization or researcher fulfilling those restrictions, provided that the owner of the BIFAP (the AEMPS) specifically authorizes the data transfer.

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
