# Peer review of "Development and Validation of Case-Finding Algorithms for Digestive Cancer in the Spanish Healthcare Database BIFAP"

_jcm, 2024, doi:10.3390/jcm13020361_

Round 1

Reviewer 1 Report

Comments and Suggestions for Authors

The manuscript has been generally well written, however minor grammatical edits are required (e.g. (a) line 279, (b) use of 'being' in multiple locations can be replaced with appropriate terms). 

Providing some statistics about the digestive cancer rates in Spain might also help in understanding the importance of the system.

One of the concerns raised in the introduction is the requirement of time and human resources. Providing more information on how the current methodology would reduce these efforts will help make the manuscript stronger as, in the current study, it appears that the reviewers were majorly involved in reviewing the data. Are there any automated algorithms set-up to avoid the extensive manual process?

Adding details in the methods section on specific terminologies used for cancer diagnosis in the text would be beneficial (e.g. lung cancer, tumor or the ICD codes only). 

The authors took additional steps to confirm the results and avoid bias by conducting sensitivity analysis. This strengthens the study.

The results have been clearly stated as per the analytical plan.

Comments on the Quality of English Language

Minor edits are required in multiple locations in the manuscript. 

Author Response

  1. The manuscript has been generally well written, however minor grammatical edits are required (e.g. (a) line 279, (b) use of 'being' in multiple locations can be replaced with appropriate terms). 

Answer: Thanks to the reviewer for making us notice. We have reviewed the whole manuscript and some grammatical corrections have been included.

  1. Providing some statistics about the digestive cancer rates in Spain might also help in understanding the importance of the system.

Answer: Thanks to the reviewer for the advice. We included a paragraph about the epidemiology of digestive cancer in Spain in the “Introduction” (lines 57-63), in the manuscript with tracked changes, as follows: “Despite remarkable advances in diagnosis and treatment, cancer remains a major challenge for public health. In Spain, the last large epidemiologic study published showed that cancer is the second cause of death after cardiovascular diseases [1]. More specifically, digestive cancer (including esophagus, gastric, hepato-biliary, pancreatic, and colorectal) is among the top-10 in number of deaths, accounting for 36% of all cancer-related deaths [1]. Further, colorectal cancer is the third most incident type of cancer among men and the second among women, resulting the most incident when both sexes are considered [1].”

  1. One of the concerns raised in the introduction is the requirement of time and human resources. Providing more information on how the current methodology would reduce these efforts will help make the manuscript stronger as, in the current study, it appears that the reviewers were majorly involved in reviewing the data. Are there any automated algorithms set-up to avoid the extensive manual process?

Answer: Thank you for bringing this to our attention. The validation process consisted in the review of the complete Electronic Health Records (EHRs) of 816 patients randomly selected. As we discuss in 2.1. Source of information, lines 132-140, in the manuscript with tracked changes, this is essential because Primary Care Physicians can modify descriptors associated with a code by adding more information or using it as a template for another diagnoses. Assessing the entire EHR allows us to distinguish valid cases from those involving patient relatives, prevalent cases, or inconclusive diagnoses when physicians are uncertain between two possibilities. The validation process, including training junior researchers, defining decision trees, the agreement step, validating the rest of the sample, conducting initial analysis, involving the Validation Committee in decision-making, fine-tuning methods, and final analysis, could extend over six months to a year of work. Automated algorithms cannot replace this extensive manual process, as, at present, they are uncapable to establish causal relationships between all information contained in the Electronic Health Records.

To emphasize the efficiency of our strategy, we included in the “Discussion”, lines 341-348, in the manuscript with tracked changes, the following: “…This was possible with controlled costs in human resources and time, as from a random sample of 816 EHRs, homogeneously clustered and manually reviewed, we achieved to extrapolate the sample validation results to the entire pool of 95,672 potential digestive cancer cases identified in BIFAP. The manual validation process, including all the stages described in the present study, took us almost a year of work. Managing huge sample sizes are common when using large electronic databases, so it is needless to say that the manual review of the whole pool of potential cancers would have been impractical, and that is the importance of designing an efficient strategy.”

  1. Adding details in the methods section on specific terminologies used for cancer diagnosis in the text would be beneficial (e.g. lung cancer, tumor or the ICD codes only). 

Answer: Thanks, totally agreed. The code list from ICPC-BIFAP and ICD-9 dictionaries included in our case-finding algorithms are shown in 2.2. Study design and development of the case-finding algorithms, lines 154-158, in the manuscript with tracked changes, as follows: “To that end, we built a sort of case-finding algorithms fitted to each type of digestive cancer that included the following ICPC-BIFAP and ICD-9 codes: for esophageal cancer: D77.2/8, 150-150.9, 230.1; for gastric cancer: D74.1, D75.1003, 151-151.9, 230.2; for pancreatic cancer: D76.1/4/5, 157-157.9, M8154/3; for hepatobiliary cancer: D77.4/5, 155-155.2, 156-156.9, 230.8, M8160/3, M8161/3, M8170/3, M8180/3, M8970/3, M8970/6; for colorectal cancer: D75.1/4/5/6/999/1004/1005, 153.-153.9, 154-154.8, 230.3-230.4”. However, as text-mining strategies are quite extensive, we prefer to detail them in the supplementary material.

It is important to highlight that ICPC-BIFAP is a dictionary modified by BIFAP to increase granularity, as it is explained in “2.1. Source of information”, lines 134-140, in the manuscript with tracked changes, so the terms shown above may differ from those of ICPC.

  1. The authors took additional steps to confirm the results and avoid bias by conducting sensitivity analysis. This strengthens the study.
  2. The results have been clearly stated as per the analytical plan.

Thanks to the reviewer for these positive comments.

Reviewer 2 Report

Comments and Suggestions for Authors

This study aimed to develop and validate algorithms for digestive cancer diagnosis using data from BIFAP. This study is complementary to previous work using similar data.

1: I highly recommend conducting a table summarizing the findings of this study in comparison to the previous studies that used similar approaches for cancer diagnosis. This will help readers better understand the merits of this study compared to others. 

2: The authors may add SIRs of BIFAB and REDECAN in the abstract section.

3: SIRs were sex-stratified, while PPVs were not. The authors might have avoided the sex-stratification of PPV because of the limited number of cases, yet they applied it in SIRs. So, the authors should consider providing the overall and sex-stratified results for both SIRs and PPVs. 

Comments on the Quality of English Language

The study is well-written. 

Author Response

This study aimed to develop and validate algorithms for digestive cancer diagnosis using data from BIFAP. This study is complementary to previous work using similar data.

1.I highly recommend conducting a table summarizing the findings of this study in comparison to the previous studies that used similar approaches for cancer diagnosis. This will help readers better understand the merits of this study compared to others. 

Answer. Thanks, we appreciate your advice. Our results are discussed and compared with the outcomes obtained from validation processes in other primary care databases (lines 349-374, in the manuscript with tracked changes.). As we state there, discrepancies identified may be attributed to the fact that some validation studies used a cancer registry as the gold standard. However, when case-findings are just based on codes without incorporating free-text, Positive Predictive Values might be lower, even if the results can be aligned with those of a cancer registry. The parameters of the case-finding algorithms differ from that used by other authors in other primary care databases. Therefore, summarizing PPVs in a table may be misleading.

No changes proposed.

2.The authors may add SIRs of BIFAB and REDECAN in the abstract section.

Answer. Thanks to the reviewer for the recommendation. However, as required by the editors we had to drastically reduce the wording of the abstract. Consequently, we only had space to show our main validation results. Nevertheless, we are open to include more information about SIRs in the abstract on editor’s demand.

The new abstract can be found in lines 36-51, in the manuscript with tracked changes.

3.SIRs were sex-stratified, while PPVs were not. The authors might have avoided the sex-stratification of PPV because of the limited number of cases, yet they applied it in SIRs. So, the authors should consider providing the overall and sex-stratified results for both SIRs and PPVs. 

Answer: Thanks to the reviewer for the comment. It is not expected to obtain differential results of PPV by gender due to the following reasons: 1) the algorithms did not include gender as a parameter, 2) the sample where PPVs were computed was randomly extracted from the pool of cancer candidates, thus representative of the distribution of gender in the population of BIFAP, and 3) there is no evidence that support that cancer records from women are of lower quality than those from men, and vice versa. However, to ensure this, we stratified the PPVs by gender and results were as follows:

Non-valid case

Valid case

Total

PPV (95%CI), %

Hepato-biliary

(n=124)

Men

12

74

86

86.7 (79.8-94.5)

Women

5

33

38

86.3 (73.4-99.2)

Esophageal

(n=152)

Men

5

134

139

96.1 (91.1-100.0)

Women

1

12

13

90.0 (85.5-94.5)

Pancreatic

(n=152)

Men

10

71

81

86.1 (77.4-94.8)

Women

5

66

71

93.4 (86.1-100.0)

Gastric

(n=158)

Men

8

92

100

94.5 (89.8-99.2)

Women

7

51

58

88.4 (78.5-98.2)

Colorectal

(n=174)

Men

6

114

120

96.3 (92.7-100.0)

Women

5

49

54

92.3 (84.5-100.0)

Overall

(n=760)

Men

38

486

524

93.2 (90.6-95.7)

Women

24

212

236

91.2 (86.9-95.4)

As shown, all PPVs remained higher than 85% regardless of gender. As expected, the number of women is lower compared to men, as the incidence of these cancer is higher among the latter in the general population. This may lead to random variation in the point estimates and confidence intervals among women.

This table was added as supplementary material (Supplementary Table S3: Stratified PPVs by sex). Moreover, it was included in the main text: “Additionally, to rule out differential quality of records, PPVs were also stratified by sex.” Lines 231-232. “Furthermore, PPVs stratified by sex remained consistent, as expected (Supplementary Table S3).” Lines 285-286, in the manuscript with tracked changes.

Round 2

Reviewer 2 Report

Comments and Suggestions for Authors

Good work. I have no more comments.